# Fault Diagnosis of Semi-Supervised Electromechanical Transmission Systems Under Imbalanced Unlabeled Sample Class Information Screening

**DOI:** 10.3390/e27020175

**Published:** 2025-02-06

**Authors:** Chaoge Wang, Pengpeng Jia, Xinyu Tian, Xiaojing Tang, Xiong Hu, Hongkun Li

**Affiliations:** 1School of Logistics Engineering, Shanghai Maritime University, Shanghai 201306, China; 2School of Mechanical Engineering, Dalian University of Technology, Dalian 116024, China

**Keywords:** semi-supervised learning, fault identification, information quantity screening mechanism, data imbalance, cost-sensitive strategy

## Abstract

In the health monitoring of electromechanical transmission systems, the collected state data typically consist of only a minimal amount of labeled data, with a vast majority remaining unlabeled. Consequently, deep learning-based diagnostic models encounter the challenge of scarcity in labeled data and abundance in unlabeled data. Traditional semi-supervised deep learning methods based on pseudo-label self-training, while alleviating the issue of labeled data scarcity to some extent, neglect the reliability of pseudo-label information, the accuracy of feature extraction from unlabeled data, and the imbalance in sample selection. To address these issues, this paper proposes a novel semi-supervised fault diagnosis method under imbalanced unlabeled sample class information screening. Firstly, an information screening mechanism for unlabeled data based on active learning is established. This mechanism discriminates based on the variability of intrinsic feature information in fault samples, accurately screening out unlabeled samples located near decision boundaries that are difficult to separate clearly. Then, combining the maximum membership degree of these unlabeled data in the classification space of the supervised model and interacting with the active learning expert system, label information is assigned to the screened unlabeled data. Secondly, a cost-sensitive function driven by data imbalance is constructed to address the class imbalance problem in unlabeled sample screening, adaptively adjusting the weights of different class samples during model training to guide the training of the supervised model. Ultimately, through dynamic optimization of the supervised model and the feature extraction capability of unlabeled samples, the recognition ability of the diagnostic model for unlabeled samples is significantly enhanced. Validation through two datasets, encompassing a total of 12 experimental scenarios, demonstrates that in scenarios with only a small amount of labeled data, the proposed method achieves a diagnostic accuracy increment exceeding 10% compared to existing typical methods, fully validating the effectiveness and superiority of the proposed method in practical applications.

## 1. Introduction

Rolling bearings play a pivotal role in the electric drive system of electric vehicles, as they not only support rotating components but also transmit torque and bear various loads, ensuring the efficient and stable operation of the vehicles. However, bearing failures can have severe consequences for electric vehicles. Minor faults may lead to reduced power output, increased energy consumption, and decreased range. In cases of severe failures, such as sudden bearing failure during high-speed travel, it may even result in electric vehicle control loss and subsequently cause traffic accidents. Therefore, conducting health condition monitoring and the diagnosis of rolling bearings holds utmost significance [1].

Deep learning (DL) technology, with its powerful data analysis and feature extraction capabilities, provides an efficient and reliable solution for the health monitoring and accurate diagnosis of rolling bearings [2,3]. However, the optimization and training of DL models heavily rely on sufficient and high-quality datasets. In the process of practical application, the time-consuming, laborious, and costly nature of manually labeling monitoring data results in a large quantity of data collected by sensors frequently remaining unlabeled [4]. Therefore, how to construct an accurate fault diagnosis model based on limited labeled data and massive unlabeled data has become a crucial issue that urgently needs to be addressed.

In recent years, semi-supervised methods have received extensive attention from scholars for their ability to integrate feature information from both labeled and unlabeled data to optimize model training [5]. The existing semi-supervised methods can be roughly divided into three categories: generative-based semi-supervised learning (SSL) methods, consistency regularization-based SSL methods, and pseudo-label-based SSL methods [6,7]. However, the varying occurrence probabilities of different health conditions in rolling bearings result in a significant class imbalance in the monitoring data [8]. This data class imbalance poses a significant challenge to the performance of semi-supervised models. To make semi-supervised models applicable to data imbalanced scenarios, scholars have proposed imbalanced fault diagnosis methods based on data preprocessing and cost-sensitive strategies [9]. Although these methods have alleviated the issues arising from data imbalance to some extent, each of them has its inherent limitations. Data preprocessing-based methods, when addressing data imbalance, may lead to issues such as model overfitting, data redundancy, and generated samples failing to capture the intrinsic characteristics of the data [10]. Moreover, cost-sensitive strategies-based methods mostly design learning mechanisms using the sample sizes of various categories, neglecting the information content of the intrinsic characteristics of the data. This can result in challenges for the model when processing data with rich decision boundary information, making it difficult to achieve effective diagnosis [11].

To solve the above problems, this paper innovatively proposes a semi-supervised deep learning (*SSDL*) fault diagnosis method for unlabeled data with information imbalance. This method not only fully utilizes limited labeled data and massive unlabeled data, but also successfully addresses the challenge of data imbalance, especially for unlabeled sample data located at the data decision boundary and rich in information content, achieving more accurate diagnosis. By introducing the proposed method, we have provided a new solution for the practical fault diagnosis of rolling bearings, which in turn provides solid and powerful support for the safe and stable operation of electric drive systems in electric vehicles.

To overcome the aforementioned challenges, this paper innovatively proposes a semi-supervised fault diagnosis technology for electric drive systems under imbalanced unlabeled sample class screening. This approach not only fully utilizes limited labeled data and vast unlabeled data but also successfully addresses the challenge of data imbalance, particularly for unlabeled sample data located at the decision boundary and rich in information content, thereby achieving precise fault diagnosis. By introducing the proposed method, we provide a novel solution for practical fault diagnosis of rolling bearings, further providing solid support for the safe and stable operation of electric vehicle drive systems. The innovations of this paper are as follows:

(1)The proposed model designs an unlabeled information screening mechanism based on active learning. This mechanism precisely selects samples located near the decision boundary that are difficult to classify clearly, based on the differences in the intrinsic feature information content of unlabeled samples. Subsequently, through a comparison of the maximum membership degrees of these unlabeled data in the classification space of a supervised model and interaction with an active learning expert system, accurate label information is assigned to them. The designed mechanism not only significantly improves the accuracy of label assignment but also effectively enhances the model’s recognition capabilities for unlabeled samples.(2)A cost-sensitive function driven by data imbalance is constructed, which can adaptively adjust the weights of different classes of samples during the model training process, thereby effectively addressing the performance bias and decreased recognition ability for minority class samples faced by the model when processing imbalanced unlabeled sample data.(3)A dynamic optimization strategy that incorporates supervised modeling and feature extraction capabilities for unlabeled samples is introduced, which leverages dynamic and iterative optimization techniques to enhance model performance while simultaneously improving the reliability of the selected unlabeled sample data.(4)Through two diverse types of bearing datasets, encompassing a comprehensive total of 12 experimental scenarios, the diagnostic performance of the proposed method is validated, providing a novel technique to tackle the issues of limited labeled samples and inadequate reliability of unsupervised models in real-world applications.

The remaining structure of this article is as follows. Section 2 introduces the related work. Section 3 provides the detailed process of the proposed ACUS-SSL model. Section 4 validates the effectiveness and superiority of the proposed model through two different datasets. Section 5 summarizes the conclusion.

## 2. Related Work

### 2.1. Semi-Supervised Deep Learning Fault Diagnosis Method

The *SSDL* methods are commonly used to address the scarcity of labeled data in traditional fault diagnosis. In SSL frameworks, autoencoder is a commonly used technique that can learn the intrinsic features of data by reconstructing unlabeled data.

However, traditional AEs exhibit limitations in terms of the stability and diversity of generated samples. To address this, Zhang et al. [12] innovated upon the Variational Autoencoder (VAE) by integrating generative and discriminative properties, thereby enhancing model stability. Despite this improvement, VAEs still face challenges in terms of the authenticity and diversity of generated samples. To further elevate the quality of generated samples, Zhang et al. [13] developed a novel Gradient Penalty Generative Adversarial Network (GAN) with multi-module learning capabilities, capable of producing high-quality samples for diagnosis. Che et al. [14] constructed a Semi-Supervised Multi-Task Convolutional Generative Adversarial Network (SM-DCGAN), which expands imbalanced samples in the generator of the network model, enabling fault diagnosis of rolling bearings under imbalanced data conditions. Nevertheless, generative SSL methods are prone to mode collapse or training instability when dealing with complex data distributions.

To address the aforementioned issues, Yu et al. [15] developed a semi-supervised method based on the principle of consistency regularization to enhance the robustness and stability of the model. Beyond the consistency regularization approach, Zhao et al. [16] introduced a semi-supervised deep sparse autoencoder method leveraging both local and nonlocal information. This method obtains topological regularization terms through intra-class compactness and inter-class discriminability, enabling fault diagnosis with unlabeled samples. Jian et al. [17] dynamically correct noisy pseudo-labels through a label updating strategy and optimize the feature consistency of the model by incorporating a consistency enhancement loss within a student–teacher network structure. Although consistency regularization methods have improved model reliability to a certain extent, the supervised model built with a limited number of labeled samples has poor credibility, leading to significant errors in the feature consistency of unlabeled samples. Consequently, it is difficult to accurately extract information from unlabeled samples.

To further explore the potential of unlabeled data, SSL methods based on pseudo-label self-training have been proposed. To fully explore the fault features contained in unlabeled data, Tang et al. [18] constructed an unsupervised network for extracting features from unlabeled data, and jointly fine-tuned it with a supervised network. By changing the way unlabeled data is utilized, they improved the accuracy of semi-supervised fault diagnosis. Sun et al. [19] proposed an unsupervised domain adaptation method based on knowledge distillation and domain-invariant feature evaluation. This method predicts pseudo-labels for the target domain through knowledge distillation and dynamically adjusts the feature alignment metric to adapt to different domains. He et al. [20] combined multi-kernel principal component analysis (MKPCA) for dimensionality reduction with pseudo-label generation to remove redundant information in the feature space and fine-tuned model parameters using pseudo-labeled data, effectively improving the classification accuracy of bearing fault diagnosis. Li et al. [21] enhanced the performance of *SSDL* models by correcting pseudo-labels using an improved Transfer Component Analysis (TCA) and K-means clustering method. Zhang et al. [12] employed a VAE to extract data features, using the features of labeled data to train a classifier and the features of unlabeled data for reconstructing the original data, thereby improving the fault diagnosis accuracy of the model.

However, although SSL methods based on pseudo-labels have alleviated the issue of insufficient labeled data to some extent, these methods struggle when dealing with data imbalance problems. Data imbalance often leads the model to prefer learning categories with a larger number of samples, thereby compromising overall performance. Therefore, how to effectively deal with data imbalance in SSL has become a critical challenge that urgently needs to be tackled.

### 2.2. Imbalance Fault Diagnosis Method Based on Data Preprocessing

To address the issue of data imbalance, scholars have proposed fault diagnosis methods for class imbalance based on data preprocessing. These methods primarily include undersampling, oversampling, and generative data processing. In terms of undersampling, Xie et al. [22] proposed a method based on progressive undersampling, which achieves uniformity in sample size and thereby solves the problem of decreased diagnostic accuracy caused by data imbalance. Zhang et al. [23] introduced an undersampling approach based on fuzzy rough sets, which ensures that information from the majority class samples is not lost while achieving accurate fault identification. Sun et al. [24] further proposed a sampling method that can adaptively adjust the undersampling rate, enabling accurate fault identification in the context of data imbalance. However, while undersampling-based methods can effectively balance the data, reducing the number of majority class samples may lead to a loss of data information, thereby affecting the generalization ability of the model.

To overcome the shortcomings of undersampling, scholars subsequently developed data preprocessing methods based on oversampling. Oversampling balances the dataset by increasing the number of minority class samples, thereby addressing the issue of data class imbalance. Hang et al. [25] proposed a MeanRadius-SMOTE method, which effectively addresses the problem of low diagnostic accuracy in imbalanced datasets by reducing the generation of useless and noisy samples. Pu et al. [26] introduced an improved SMOTE oversampling algorithm for fault diagnosis in the context of highly imbalanced data. Duan et al. [27] utilized GAN as an oversampling tool to compensate for imbalanced datasets in diagnostic tasks. However, while oversampling-based methods balance the data classes, they undoubtedly significantly increase the number of samples, thereby increasing the computational time of the model. Additionally, oversampling can easily lead to overfitting of the model to the training data, thereby reducing the diagnostic performance of the model.

Furthermore, to compensate for the shortcomings of oversampling methods, generative data processing approaches have emerged, which achieve data balance by generating high-quality minority class samples. Fei et al. [28] proposed a diagnostic method based on time-frequency DCGAN processing, which achieves uniformity in sample quantity by expanding fault samples. Zhao et al. [29] integrated the VAE with CNN, solving the problem of unbalanced sample numbers by utilizing the VAE to generate artificial signals. Li et al. [30] built a Variational Autoencoder Generative Adversarial Network (VAE-GAN) framework, which augments samples of time-frequency images to achieve fault diagnosis under imbalanced conditions. However, generative data processing methods may not fully capture the useful feature information in real data when generating minority class samples, resulting in discrepancies in the distribution between the generated samples and real data. This, in turn, affects the accuracy and reliability of the diagnostic model.

Given the limitations of the aforementioned methods, data imbalance diagnosis approaches based on cost-sensitive functions have emerged as a promising research direction. These methods introduce a cost-sensitive mechanism to enhance the classification performance of minority class samples, providing a new perspective for addressing data imbalance issues.

### 2.3. Imbalance Fault Diagnosis Method Based on Cost-Sensitive Strategy

In recent years, scholars have actively explored and proposed various data imbalance fault diagnosis methods based on cost-sensitive strategies. Jiang et al. [31] proposed an SVM classifier based on cost sensitivity, which adaptively obtains sample weight factors, enhancing diagnostic accuracy while maintaining the simplicity of the model structure. Subsequently, Deng et al. [32] constructed a knowledge distillation-oriented cost-sensitive ensemble learning framework, which significantly improved diagnostic performance under extreme data imbalance by increasing attention to minority class samples. Tang et al. [33] designed a cost-sensitive LightGBM fault detection framework, further enhancing the accuracy of fault identification. However, when dealing with large-scale data, the feature extraction capabilities of the shallow networks employed in these methods appear insufficient. To address this issue, Ren et al. [34] effectively suppressed the dominance of majority class samples by combining sample quantity distribution, class convergence trends, and sample convergence trends, thereby enhancing classification performance. However, this method still relies on the proportion of each class in the total sample size for class weight allocation, failing to achieve adaptive updates. To overcome this limitation, Zhang et al. [35] designed a novel cost-sensitive learning strategy based on policy gradients and reinforcement learning, known as the Actor–Critic framework, which enables dynamic adjustment of weights. Yang et al. [36] combined a multi-scale feature focusing network with an adaptive cost-sensitive loss strategy, solving the problem of the network model overfocusing on majority class samples by adaptively adjusting the cost weight matrix. Zhao et al. [37] provided a practical solution for data imbalance in fault diagnosis by fine-tuning the cost-sensitive matrix through adaptive search for hyperparameter combinations. Additionally, Mao et al. [38] designed the Transferable Dynamic Enhanced Cost-Sensitive Network (TDECN), which successfully addressed fault diagnosis challenges under extreme data imbalance by enhancing attention to minor or minority class samples. Despite these methods being able to dynamically generate sample weights and address data imbalance issues, the accuracy of class weight allocation still needs improvement when dealing with complex multi-class classification tasks. Subsequently, Xu et al. [39] successfully resolved the decline in classification accuracy caused by inconsistent class attention in traditional classifiers through innovative design of cost-sensitive strategies. Peng et al. [40] proposed the Non-Revisiting Genetic Cost-Sensitive Sparse Autoencoder, which achieved a certain degree of balance in multi-class classification tasks through reasonable allocation of class weight coefficients, further enhancing the applicability of this method in various classification tasks.

However, it is noteworthy that most of the aforementioned methods for addressing data imbalance issues are implemented under supervised models. Furthermore, when setting misclassification penalties, existing methods primarily rely on the ratio between classes or the proportion of each class’ samples to the total samples, which ignores the variability in the amount of information between samples. Given the significance of information content in fault diagnosis, it is imperative to devise more sophisticated cost-sensitive strategies tailored to the variability in information content among samples, thereby further enhancing the accuracy of fault diagnosis.

## 3. Semi-Supervised Fault Diagnosis Method Under Imbalanced Unlabeled Sample Class Information Screening

To effectively leverage these unlabeled data, traditional semi-supervised methods based on pseudo-label self-training have emerged, aiming to optimize supervised diagnostic models built from a limited amount of labeled data through a vast quantity of unlabeled data. However, these methods often overlook a crucial issue: the imbalance of sample data within the unlabeled data. This imbalance can cause the model to preferentially focus on categories with a larger number of samples during training, potentially leading to a decline in the recognition ability for minority categories. Consequently, traditional SSL methods may exhibit performance limitations and biases when confronted with the imbalance of information in unlabeled data.

To effectively address the above issue, this paper proposes a novel semi-supervised fault diagnosis method under the scenario of imbalanced class information screening in unlabeled samples. Specifically, an information screening mechanism for unlabeled data based on active learning is first designed. This mechanism discriminates based on the differences in the intrinsic feature information content of fault samples, accurately screening out unlabeled data that are located near decision boundaries and difficult to clearly separate. Subsequently, by combining the maximum membership degrees of these unlabeled data in the classification space of the supervised model and comparing them with an active learning expert system, label information is assigned to the screened unlabeled data. Furthermore, a cost-sensitive function driven by data imbalance is constructed to address the information imbalance issue in unlabeled data screening, adaptively adjusting the weights of different class samples during model training, thereby guiding the training of the supervised model. Ultimately, through dynamic optimization of the supervised model and the feature extraction capability of unlabeled samples, the recognition ability of the diagnostic model for unlabeled samples is significantly enhanced. The flowchart of the proposed method is shown in Figure 1.

### 3.1. Active Learning-Based Mechanism for Unlabeled Sample Information Screening

Traditional pseudo-labeling methods for unlabeled data struggle to generate reliable pseudo-labels due to their dependence on inaccurate supervised models. This can result in erroneous label information, thereby misleading the learning direction of the supervised model. To tackle this problem, this paper proposes an information filtering mechanism for unlabeled data based on active learning. The specific process is outlined as follows.

**Step 1:** Feature extraction of unlabeled data based on supervised model.

During the model training process, labeled and unlabeled data jointly constitute the sample set for model training and selection, which can be expressed as follows:(1)$$X=[x_{l},x_{u}]$$
where $x_{l}$ represents the labeled data used for model training, and $x_{u}$ denotes the unlabeled data used as the screening target for the supervised model. The training process of a supervised model can be divided into forward propagation and backward optimization phases.(2)$$\hat{y}=F_{\sup\_m}(Net_{l},Tr_{global},x_{l})$$(3)$${\mathrm{loss}}_{\sup\_m}=\frac{1}{\mathrm{nl}}\sum_{n=1}^{nl}y_{n}\log{\hat{y}}_{n}$$(4)$$\nabla loss_{\sup\_m}(Tr_{global})=\frac{1}{N}(y-\hat{y})\sigma^{\prime }(Tr_{global})$$(5)$$Tr_{global}=Tr_{global}-\mathrm{lr}\times \nabla loss_{\sup\_m}(Tr_{global})$$
where $F_{\sup\_m}$ represents the supervised model forward propagation function, ${\mathrm{Net}}_{l}$ is the network structure of supervised model, $Tr_{global}$ denotes the global model parameters of supervised model, ${\hat{y}}_{n}$ represents the predicted label, $loss_{\sup\_m}$ represents the average forward propagation error loss of supervised model, and nl represents the labeled sample size. σ denotes the model’s activation function during the information propagation process. lr represents the step size for updating model parameters. After the initial training of model, the trained global model parameters $Trained_{global}$ are saved and used to extract features of unlabeled data in the classification space of the supervised network. The expression for this is as follows [41]:(6)$${Feature}_{class}=F_{\sup\_m}({Net}_{l},Trained_{global},x_{u})$$
where $Feature_{class}$ denotes the classification features of unlabeled samples, and these classification features drive the calculation of the information content of unlabeled data. In the field of semi-supervised fault diagnosis, due to the common reliance on only a small amount of labeled data for training, the performance of supervised models is often unreliable. Therefore, from the perspective of maximizing information utilization, extracting feature information from unlabeled data and predicting label information in the classification space becomes crucial. Furthermore, obtaining such feature and label information typically enhances the feature extraction capability of supervised models. Equation (6) represents the utilization of the model parameter information $Trained_{global}$ contained within a trained supervised model with a structure of $Net_{l}$, to extract feature information from unlabeled data $x_{u}$ layer by layer, thereby obtaining category features $Feature_{class}$ in the classification space.

**Step 2:** Evaluation of information content in unlabeled data.

In classification tasks, we pay special attention to those fault samples that are difficult to clearly distinguish, as they are crucial for learning a more precise decision boundary. This paper screens unlabeled sample data with high information content near the decision boundary by designing an information content evaluation mechanism. The process of information content evaluation is as follows:(7)$$diff=-\sum_{c=1}^{C}Feature_{class_{c}}\log_{2}Feature_{class_{c}}$$(8)$$diff_{norm}=\frac{diff}{\log_{2}C}$$
where C denotes the total number of categories. Equation (7) is used to characterize the average information content of unlabeled data belonging to each category in the classification space. The smaller the average information content, the easier it is to discriminate the category it belongs to. Conversely, the larger the average information content, the more difficult it is to determine the category of the unlabeled data [42]. To process the sample data more effectively, Equation (8) maps the value range of the average information content to [0, 1], thereby facilitating the subsequent sample selection process. Subsequently, using Equations (7) and (8), we can screen out samples with rich information content.

**Step 3:** Unsupervised data filtering based on information content discriminability.

After screening out the nu groups of unlabeled samples with the highest $D{\mathrm{iff}}_{norm}$ values in Step 2, we calculate the maximum membership degree of each sample in the classification space, as shown in Equation (9):(9)$$\begin{array}{c} {\mathrm{infox}}_{u,1}=\max(Featuree_{x_{u,1}},class)\\ \begin{array}{c} {\mathrm{infox}}_{u,2}=\max(Featuree_{x_{u,2}},class)\\ \vdots \end{array}\\ {\mathrm{infox}}_{u,nu}=\max(Featuree_{x_{u,nu}},class) \end{array}$$
where ${\mathrm{infox}}_{u,nu}$ represents the maximum membership degree of the selected unlabeled samples in the classification space. Subsequently, the obtained maximum membership degrees are input into the scoring function within the selection mechanism, as illustrated by the following formula:(10)$$\rho_{i}=score({\mathrm{infox}}_{u,1},{\mathrm{infox}}_{u,2},\cdots ,{\mathrm{infox}}_{u,nu})$$(11)$$x_{u,\mathrm{screering}}=\mathrm{screering}(\rho_{1},\rho_{2},\cdots ,\rho_{nu})$$
where $\rho_{i}$ denotes the score of the unlabeled data, and $x_{u,screering}$ represents the unlabeled samples selected based on their scores within the selection mechanism. Following this, the samples with high information content are filtered out and submitted to an expert system for annotation. After annotation, these samples are expanded into the original labeled sample set. Such an approach not only enhances the accuracy and reliability of the data but also introduces more high-quality and representative samples into the model training, thereby further improving the classification performance of the model.

### 3.2. Cost-Sensitive Strategy to Address Category Imbalance in Unlabeled Sample Screening

Although expanding the original labeled dataset with unlabeled samples screened in Section 3.1 can effectively enrich the sample information, the selected samples often suffer from class imbalance. Directly incorporating these imbalanced samples into the model for training will cause the model to drift towards the majority class, resulting in a decline in its recognition capability for minority class samples. To address this issue, this paper proposes a cost-sensitive strategy to handle the class imbalance problem in unlabeled sample screening. The specific steps are as follows:

**Step 1:** Calculate the information content of the augmented labeled samples.

The augmented labeled sample set can be represented as follows:(12)$$x_{ex\_l}=[x_{l},x_{u,screering}]$$

The information content of these samples is recalculated using a supervised model, as shown below:(13)$$diffx_{ex\_l,n}=\frac{-\sum_{nu=1}^{N}F_{\sup\_m}(Net_{l},Trained_{global,x_{ex\_l,nu}})\log_{2}F_{\sup\_m}(Net_{l},Trained_{global,x_{ex\_l,nu}})}{\log_{2}C}$$

**Step 2:** Develop an information content-driven cost-sensitive strategy.

To mitigate the impact of imbalanced data on the performance of supervised models, a new cost-sensitive loss for the model is designed using the information content of samples with various labels, as shown in Equation (14),(14)$$loss_{x_{ex-l}}=\sum_{n=1}^{nl}loss_{x_{l,n}}+\sum_{nu=1}^{N}loss_{x_{u,screening,nu}}$$
where ${\mathrm{loss}}_{x_{l,n}}$ and ${\mathrm{loss}}_{x_{u,screering,\mathrm{nu}}}$ represent the loss for individual labeled samples and selected unlabeled samples, respectively, as shown in Equations (15) and (16).(15)$${loss}_{x_{l,n}}=y_{l,n}\log{\hat{y}}_{l,n}$$(16)$$loss_{x_{u,screening,nu}}={\mathrm{diff}}_{x_{ex,l,u}}\times {\hat{y}}_{u,screening,nu}\log{\hat{y}}_{u,screening,nu}$$

Traditional cost-sensitive learning strategies often append a cost-sensitive matrix to the classification loss function, where the matrix sets misclassification penalty coefficients based on the number or proportion of samples in each category. However, this approach may not accurately reflect the actual class distribution characteristics of the sample data, thereby weakening the effectiveness of cost-sensitive learning. In the model loss function designed in Equation (16), we comprehensively consider two factors. Firstly, to alleviate the adverse effects of the screened imbalanced samples on supervised model training, we retain the contribution of the loss from the original balanced samples ${\mathrm{loss}}_{x_{l,n}}$ during the training process. Secondly, for each sample, we determine the classification penalty term ${\mathrm{loss}}_{x_{u,screering,\mathrm{nu}}}$ based on the amount of information contained in the selected unlabeled samples. This approach aims to enable the model to more accurately determine the misclassification cost of each sample during the learning process, rather than merely relying on the class proportion of the samples.

### 3.3. Dynamic Optimization Strategy for Supervised Model and Unlabeled Data Feature Extraction Capability

The performance of DNNs heavily relies on the quantity and quality of training data. Therefore, our objective is to effectively leverage filtered unlabeled data to enhance the performance of diagnostic models. The complementary relationship between supervised models and the feature extraction capability of unlabeled data offers potential for achieving this goal. Specifically, we incorporate the filtered, information-rich unlabeled samples into the original labeled dataset for further training of the supervised model, thereby improving its feature extraction capability for unlabeled samples. Meanwhile, through this dynamic iterative optimization approach, not only is the model’s performance enhanced, but the reliability of the filtered unlabeled samples is also improved. In summary, the dynamic optimization process for supervised models and the feature extraction capability from unlabeled data is implemented as follows:

**Step 1:** Optimize supervised model utilizing filtered unlabeled sample information.

In this step, we adopt a cost-sensitive loss function to drive the model training and leverage reverse dynamic optimization techniques to enhance the reliability of model parameters.(17)$$Tr_{global}=Tr_{global}-\mathrm{lr}\nabla loss(Tr_{global})$$(18)$$\nabla loss(Tr_{global})=\nabla loss_{x_{l}}(Tr_{global})+\nabla loss_{x_{u,screening}}(Tr_{global})$$

Through multiple rounds of training using Equations (17) and (18), we continuously adjust the model parameters, thereby improving the model’s recognition capabilities and feature extraction accuracy for samples. Furthermore, during the reverse optimization process, the model’s misclassification penalty is driven by the information content of the samples; namely, samples with greater information content incur larger penalties for misclassification, which helps the model to focus more on those samples that contribute significantly to performance improvement. The specific formulas are as follows:(19)$$\nabla loss_{x_{l}}(Tr_{global})=\frac{1}{nl}(y_{x_{l}}-{\hat{y}}_{x_{l}})\sigma^{\prime }(Tr_{global})$$(20)$$\nabla loss_{x_{u,screening}}(Tr_{global})=diff_{x_{ex\_l,n}}\frac{1}{N}(y_{x_{u,screening,n}}-{\hat{y}}_{x_{u,screening,n}})\sigma^{\prime }(Tr_{global})$$

Therefore, as indicated by Equation (20), compared to traditional cost-sensitive loss functions, the constructed imbalanced sample information-driven cost-sensitive function incorporates an additional misclassification penalty coefficient based on the information content of the samples ${\mathrm{diff}}_{x_{ex,n}}$. This design makes the model more sensitive to classification errors for samples with higher information content during the training process. Thus, when these important samples are misclassified, it triggers a more significant update of the model parameters. Such a mechanism enables the diagnostic model to better adapt to and fit the distribution characteristics of imbalanced class sample data, thereby effectively improving the accuracy of sample classification.

**Step 2:** Model iteration optimization and high-quality unlabeled sample screening.

Step 1 and Step 2 form a cyclic iterative optimization process. In each iteration, new sample information is incorporated into the model training process, leading to continuous optimization of the model parameters. The optimized model parameters ${\mathrm{Tr}}_{global}$ aid in selecting the most reliable unlabeled samples and effectively incorporating them into the training of the supervised model, thereby achieving continuous enhancement of model performance. This process can be mathematically expressed as follows:(21)$$Feature_{class}^{ex}=F_{\sup\_m}^{ex}(Net_{l}^{\mathrm{ex}},Trained_{global}^{ex},x_{u}^{re})$$
where $Feature_{class}^{ex}$ represents the classification spatial features extracted from the remaining unlabeled data $x_{u}^{re}$ by a model retrained after expanding the sample set ${\mathrm{Net}}_{l}^{ex}$. The retrained model can effectively separate samples that were previously difficult to filter and located near the decision boundary during previous rounds of model training. Through this approach, unlabeled samples with higher remaining information content can be selected, thereby further improving the accuracy of assessing the information content of unlabeled samples, as shown in Equation (22).(22)$${diffx}_{u}^{ex}=\frac{-\sum_{n=1}^{N}F_{\sup\_m}^{ex}({Net}_{l}^{ex},Trained_{global,x_{ex\_l,n}}^{ex},x_{u}^{re})\log_{2}F_{\sup\_m}^{ex}({Net}_{l}^{ex},Trained_{global}^{ex},x_{u}^{re})}{\log_{2}C}$$

## 4. Experiments and Analysis

In this section, the Case Western Reserve University (CWRU) bearing dataset [43] and the Shanghai Maritime University (SMU) bearing dataset are used to verify the proposed approach. This section presents specific experimental setups and analyses to comprehensively validate the effectiveness and practicality of the proposed model.

### 4.1. Experimental Data Description

#### 4.1.1. CWRU Bearing Dataset

The CWRU rolling bearing experimental platform, as illustrated in Figure 2, primarily consists of key components such as a drive motor, torque transducer and encoder, and a load motor. To simulate different types of faults, electrical discharge machining technology is utilized to artificially implant 0.007-inch pitch faults on the inner race, outer race, and rolling elements of the bearings, respectively. During the experiment, the faulty bearing under test is installed on the drive end of the motor, using SKF6205 bearing with parameters in Table 1. During data acquisition, the motor speed was maintained at a constant 1772 rpm with a constant load of 1 Hp applied. To precisely capture the vibration signals resulting from bearing faults, accelerometers were installed on both the drive end and fan end of the motor, and the data sampling frequency was set to 12 KHz. Table 2 provides the types of experimental bearing faults. The collected vibration data of bearings under different health states are shown in Figure 3.

#### 4.1.2. SMU Bearing Dataset

To further validate the effectiveness and advantage of our proposed method, we conducted experiments on an SMU rolling bearing fault diagnosis platform, as illustrated in Figure 4, which primarily comprises a drive motor, a frequency converter, a bearing housing, a hydraulic loading device, vibration acceleration sensors, and a data collection device. The sensors are linked to an industrial computer installed with DHDAS software through an NI PCI-4472 card, thereby collecting experimental data. The tested bearing is an ER-16K deep groove ball bearing, and its parameters are shown in Table 3. The experiment simulates a total of ten health conditions of the rolling bearing, including normal condition, inner race pitting 0.2 (I0.2), inner race pitting 0.4 (I0.4), inner race pitting 0.6 (I0.6), outer race pitting 0.2 (O0.2), outer race pitting 0.4 (O0.4), outer race pitting 0.6 (O0.6), ball pitting 0.2 (B0.2), ball pitting 0.4 (B0.4), and ball pitting 0.6 (B0.6), as listed in Table 4. During the experiment, the motor speed is fixed at 1800 rpm, and the load on the magnetic particle brake is set to 15 Nm. The vibration signals are accurately collected with a sampling frequency set at 25,600 Hz for each condition, and the vibration data of the bearings are described in Figure 5.

### 4.2. Analysis of Experimental Results

#### 4.2.1. CWRU Dataset Analysis

In this section, experimental scenarios are designed utilizing a CRWU bearing dataset. When dividing samples, a sliding window with a length of 400 is employed to intercept the bearing data. Furthermore, to meet the input requirements of the CNN, each intercepted sample is transformed into 20 × 20 two-dimensional data. After constructing the sample dataset, multiple experimental scenarios are set up to validate the performance of various model algorithms. The comparison methods are described in Table 5.

The multiple experimental scenarios designed are shown in Table 6. It can be observed that in Experimental Scenario 1, the number of labeled samples is 4 × 10, while the number of unlabeled sample data is 4000, 6000, and 8000 respectively, with the screened number of unlabeled samples being 50, 100, and 200 respectively. In Experimental Scenario 2, the number of labeled samples is reduced to 4 × 5, while the number of unlabeled samples and the screened number of unlabeled samples remain consistent with those in Scenario 1.

Firstly, the performance of various models is tested in Experiment Scenario 1. Specifically, the CNN-based method aims to validate the performance of the baseline model without leveraging unlabeled sample information. Subsequently, by adjusting the number of labeled samples and gradually incorporating unlabeled sample information, we investigate the performance trends of the π-model and VAE-SSL model. Furthermore, to illustrate the specific impact of selecting different quantities of unlabeled samples on the performance of CNN-SSL, AUS-SSL, and ACUS-SSL models under varying sizes of unlabeled datasets, corresponding comparative experiments are conducted. The model architecture parameters for the proposed method are presented in Table 7.

Under Experiment Scenario 1, the diagnostic results of the comparison methods and the proposed method are presented in Table 8. As evident from Table 8, it is difficult to obtain an effective diagnostic model by training a CNN with only 10 labeled samples per category without utilizing any unlabeled samples. However, by comparing the experimental results from the second to the fourth rows, it can be observed that, with the number of labeled samples remaining constant, increasing the number of unlabeled samples can improve the diagnostic accuracy of various semi-supervised models. This is because a large number of unlabeled samples provide rich fault feature information for these models, thereby enhancing the reliability of model feature extraction.

Comparing the experimental results in the fourth and fifth columns of the second row, it is found that the π-model, by introducing feature consistency loss as a regularization term, can effectively extract fault-related feature information from unlabeled samples, resulting in an improvement in performance compared to the CNN model trained solely on labeled samples. Compared to the π-model, the CNN-SSL model adopts a different strategy. It applies a screening mechanism to the predictions of unlabeled samples to obtain relatively reliable pseudo-labeled data, leading to a certain improvement in diagnostic accuracy. However, it is worth noting that the CNN-SSL model finds it difficult to ensure the complete reliability of the unlabeled data when screening pseudo-labels. Additionally, even if relatively reliable pseudo-labels can be obtained through strict screening mechanisms, the strictness of the screening conditions often makes it difficult to obtain a large number of pseudo-labels, thereby affecting the reliability of the model.

Furthermore, comparing the experimental results in the fifth and sixth columns of the second row reveals that the diagnostic accuracy of VAE-SSL is higher than that of CNN-SLL. This is because VAE-SSL employs a generative semi-supervised strategy, which balances the data distribution of different categories during the training process, thereby reducing the impact of data imbalance on diagnostic accuracy. This capability enables VAE-SSL to extract and represent fault features more accurately than CNN-SSL. However, comparing the sixth and seventh columns of the second row shows that the performance of VAE-SSL is lower than that of AUS-SSL. This is because the information-based screening mechanism in the AUS-SSL model can effectively filter out samples rich in classification information from unlabeled samples. These filtered samples, after interacting with an expert system, can obtain fully reliable label information. Subsequently, incorporating this reliable label information into the model training process can improve the diagnostic performance of the model.

Comparing the experimental results in the seventh and eighth columns of the second row, it is evident that the proposed ACUS-SSL method achieves the highest diagnostic accuracy. This is because, while AUS-SSL ensures the reliability of unlabeled information screening, it ignores the imbalance of unlabeled screening information, leading the model to bias towards the majority class and compromising its ability to recognize minority class samples. However, the proposed ACUS-SSL introduces an imbalance-information-driven cost-sensitive strategy, ensuring the reliability of the model and demonstrating satisfactory results. Specifically, with only 50 unlabeled samples screened at an imbalance ratio of 1:3:4:2, the model achieves a diagnostic accuracy of 93.75%, representing a significant improvement of 13.75% in diagnostic accuracy compared to the AUS-SSL model.

Additionally, from the results of Experiments 2 to 3, it can be observed that although we increased the number of screened unlabeled samples, the imbalance ratio of the samples also continued to rise. However, it is noteworthy that our proposed ACUS-SSL method can still effectively overcome this challenge and achieve high diagnostic accuracy.

The confusion matrix for Experiment 1 is illustrated in Figure 6. Observing Figure 6, it is evident that the proposed ACUS-SSL method achieves precise identification for each fault category, with fault discrimination accuracy higher than other comparative methods. This is attributed to its reliable unlabeled sample information screening mechanism, which accurately selects high-quality unlabeled data to effectively augment the original labeled training dataset, thereby enhancing the model’s diagnostic capability. Additionally, by constructing an imbalance-driven cost-sensitive function, the method addresses the potential information imbalance issue encountered during unlabeled sample screening, ensuring the recognition accuracy of the model for minority fault categories while maintaining the overall fault diagnosis performance of the model.

To validate the diagnostic capability of the proposed method under extreme conditions, especially when only a very limited number of original labeled samples are available, we specifically designed Experiments 4 to 6 in Scenario 2. The experimental results are presented in Table 9.

As evident from Table 9, the performance of each diagnostic model decreases as the number of labeled samples decreases. This is because the reliability of supervised models trained with a smaller number of labeled samples is greatly compromised, which in turn affects the credibility of the unlabeled data features extracted from these models. Specifically, by comparing the data in Row 2 of Table 8 and Table 9, we can observe that the unreliability of supervised models directly reduces the effectiveness of unlabeled samples in enhancing model performance, thereby affecting the performance of different semi-supervised models. Furthermore, by comparing the results of Experiment 5 and Experiment 6, it is evident that, under the condition of ensuring reliable unlabeled sample selection, incorporating the selected unlabeled samples with rich information into the training of supervised models can significantly improve the accuracy of sample recognition. Moreover, the proposed ACUS-SSL method consistently outperforms other comparative methods in diagnosing various faults. This fully demonstrates the effectiveness of the dynamic optimization strategy for supervised models and unlabeled sample feature extraction capabilities designed in Section 3.3. Additionally, it underscores the superiority of the proposed method in screening unlabeled sample class information and handling imbalanced data scenarios.

The confusion matrix for Experiment 4 is shown in Figure 7. It is clearly observable that, even when faced with the challenge of an extremely limited number of labeled samples, which results in a relatively low reliability of the constructed supervised model, our proposed ACUS-SSL method is still able to accurately identify each type of fault. This performance demonstrates the robustness of the ACUS-SSL method, namely, its ability to maintain high robustness with very few labeled samples. This provides a new solution to address the scarcity of labeled samples and model reliability issues in practical applications.

#### 4.2.2. SMU Dataset Analysis

The SMU bearing dataset differs significantly from the CWRU dataset in terms of experimental design, fault types, and data sampling. In particular, the SMU bearing data not only covers multiple fault types of bearings but also includes varying degrees of damage, providing a more comprehensive testing platform for the validation of various diagnostic algorithms.

Firstly, for the vibration data of bearings monitored under different health states, a sliding window with a length of 400 is used to extract samples. Secondly, to meet the input requirements of the CNN, each extracted sample is transformed into 20 × 20 two-dimensional data. Finally, multiple experimental scenarios with the same number of labeled samples but different numbers of unlabeled samples are constructed to verify the performance of various diagnostic models. Specifically, in Experimental Scenario 3, the number of labeled samples is 10 × 10, and the number of unlabeled samples is 10,000, 15,000, and 20,000, respectively, with the selected number of unlabeled samples being 50, 100, and 200. In Experimental Scenario 4, the number of labeled samples is reduced to 10 × 5, while the number of unlabeled samples and the selected number of unlabeled samples remain the same as in Scenario 3. The detailed experimental scenario settings are shown in Table 10.

The detailed test results for Experiments 7 to 9 in Scenario 3 are listed in Table 11. Observing Table 11, it is evident that traditional CNNs struggle to build a reliable fault diagnosis model when only a very limited number of labeled samples are available. The reason lies in the fact that when the diagnostic task requires not only accurate differentiation of fault types but also further subclassification of fault severity, a large amount of sample data is needed to support the learning and formation of clear decision boundaries. Comparing the diagnostic results of CNN and π-mode, it can be seen that when the selected unlabeled samples are added, the π-model method enhances the model’s feature extraction capability by utilizing the feature information of unlabeled data, resulting in higher fault diagnosis accuracy than CNN. Furthermore, comparing the diagnostic results of π-model with CNN-SSL, VAE-SSL, and AUS-SSL, it is apparent that these methods outperform π-model in diagnostic performance. This is attributed to their use of self-training to assign pseudo-labels to unlabeled data, thereby enriching the sample feature information required for supervised model training and enhancing the model’s diagnostic performance. However, it is worth noting that despite these improvements in performance, the accuracy of fault diagnosis is still not ideal. This is primarily due to the issue of class imbalance in the selected unlabeled samples, which adversely affects the learning direction of model parameters and thus constrains the further improvement of the model’s fault diagnosis performance. Fortunately, our proposed ACUS-SSL method effectively utilizes a cost-sensitive function designed for imbalanced information to guide and correct the learning direction of model parameters, thereby significantly enhancing the model’s fault recognition performance. Therefore, ACUS-SSL maintains excellent diagnostic capability even when faced with imbalanced sample data.

Finally, when comparing the overall results of Experiments 7 to 9, we can observe that after introducing a significant number of unlabeled samples to enhance the models’ feature capture capabilities, the π-model, CNN-SSL, VAE-SSL, and AUS-SSL all demonstrate improvements in fault diagnosis accuracy. This phenomenon underscores the significance of unlabeled samples in augmenting the models’ fault feature extraction and recognition capabilities. By effectively leveraging these unlabeled samples, these methods not only enrich the training data but also further enhance the models’ sensitivity and precision in detecting intricate fault features. Furthermore, the proposed ACUS-SSL method consistently exhibits the highest diagnostic accuracy. This is because when facing complex diagnostic tasks, the proposed method ensures the reliability of the label information of the selected unlabeled samples on the one hand. On the other hand, by constructing a cost-sensitive function driven by data imbalance, it effectively addresses the issue of information imbalance in unlabeled sample selection, thereby adaptively adjusting the weights of different classes of samples during model training and improving the model’s recognition ability for minority or difficult-to-classify categories.

Figure 8 presents the confusion matrices of various diagnostic models in Experiment 7. As observed from Figure 8, the proposed ACUS-SSL method achieves precise identification of each fault category in complex fault diagnosis tasks.

To validate the applicability of the proposed method in scenarios with fewer labeled samples, we designed Experiments 10 to 12 in Scenario 4 for comparison with Experiments 7 and 9. The diagnostic results of each model method are presented in Table 12. As can be seen from Table 12, even under the severe challenge of an extremely limited number of labeled samples, the proposed method still demonstrates superior performance compared to other comparison methods. This significant advantage not only verifies the strong decision-making and recognition capabilities of the proposed method but also highlights its ability to accurately mine distinguishable fault features from limited samples, which is consistent with the conclusions drawn from Table 9.

Furthermore, when comprehensively comparing the analysis results of a total of 12 experimental scenarios on two different datasets, we can conclude that the presented method exhibits more outstanding diagnostic performance compared to other methods, demonstrating a strong advantage in complex diagnostic tasks.

Figure 9 displays the confusion matrices of various diagnostic models in Experiment 10. It is evident from Figure 9 that, even with extremely limited labeled samples, the proposed method can accurately obtain high-quality unlabeled data by selecting informative unlabeled samples. Additionally, through the designed cost-sensitive strategy, the model can accurately capture fault features in an imbalanced data environment, thereby achieving precise identification of each fault category.

Furthermore, to highlight the outstanding performance of the proposed method across different experimental scenarios, Figure 10 intuitively displays the comparison of accuracy among various diagnostic models in all 12 experimental scenarios. It can be observed directly from the figure that the diagnostic performance of the proposed method is superior to other comparative methods in every experimental scenario. Therefore, when faced with the challenge of relatively scarce labeled data, the proposed method can efficiently utilize a large amount of unlabeled data, thereby effectively enhancing the fault diagnosis performance of the model. Not only does the proposed method reduce the dependence on a large amount of labeled data, but it also significantly improves the accuracy and reliability of fault diagnosis. This provides a novel solution and approach for addressing the challenges of scarce labeled samples and imbalanced semi-supervised fault diagnosis in practical industrial applications.

## 5. Conclusions

Addressing the scarcity of labeled samples in condition monitoring of electromechanical drive systems and the challenges faced by *SSDL* models, such as inaccurate feature extraction, low reliability of pseudo-labels, and imbalanced class distribution of unlabeled samples, this paper proposes a novel semi-supervised diagnostic method under imbalanced unlabeled sample class information screening for the precise fault identification of critical components in electromechanical drive systems. The proposed method particularly focuses on enhancing the accuracy of diagnostic models in environments with imbalanced unlabeled sample class information screening. In the proposed approach, a mechanism for unlabeled data screening based on active learning is introduced. This mechanism accurately screens out samples located near decision boundaries and with high classification difficulty, based on the variability of intrinsic feature information in unlabeled samples. Subsequently, by interacting and comparing the maximum membership degree of these unlabeled data in the classification space of the supervised model with an active learning expert system, accurate label information is assigned to these samples, significantly improving the precision of label assignment. Furthermore, to address the issue of class information imbalance during the screening of unlabeled samples, we have constructed a cost-sensitive function driven by data imbalance. This function ensures the recognition accuracy of the model for faults in minority classes, thereby enhancing the reliability of the model’s diagnosis. Additionally, we have introduced a dynamic optimization strategy for the supervised model and the feature extraction capability of unlabeled samples. This strategy not only enhances the overall diagnostic performance of the model but also further improves the precision of the screened unlabeled sample data. Validation through two different datasets encompassing a total of 12 experimental scenarios demonstrates the effectiveness of the proposed method. Even in scenarios with extremely limited labeled samples, the method can still accurately identify each type of fault, and its diagnostic accuracy is significantly higher than that of existing comparison methods. Therefore, the method proposed in this paper provides a new and effective solution to address the scarcity of labeled samples and the insufficient reliability of unsupervised models in practical applications.

This study primarily focuses on the impact of unbalanced unlabeled data on model performance in SSL scenarios where labeled samples are scarce. In future research, we will devise new DL strategies to address the negative effects of data heterogeneity on model learning, specifically targeting the variability in operating conditions and low quality of the data.

## Figures and Tables

**Figure 1 entropy-27-00175-f001:**
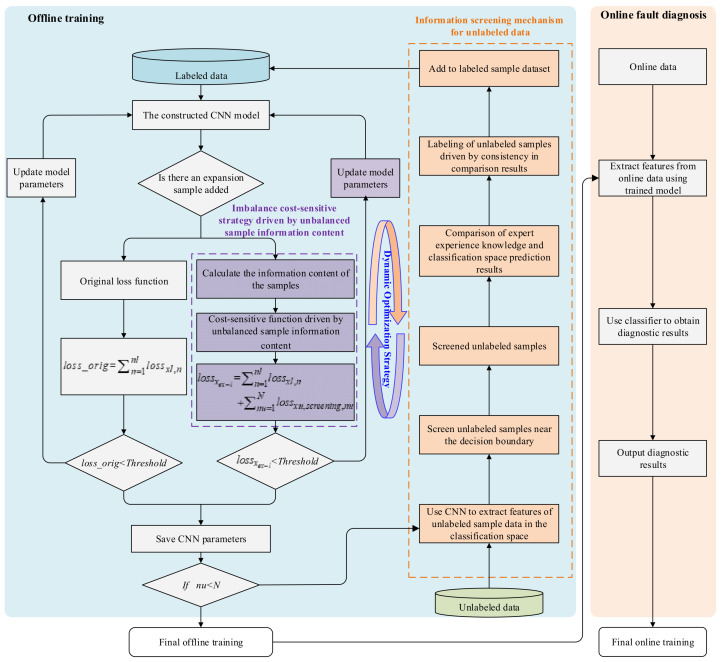
Flowchart of the proposed ACUS-SSL diagnostic model.

**Figure 2 entropy-27-00175-f002:**
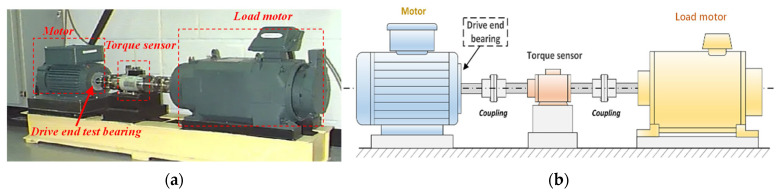
CWRU experimental platform: (**a**) bearing testing system and (**b**) schematic diagram of the platform.

**Figure 3 entropy-27-00175-f003:**
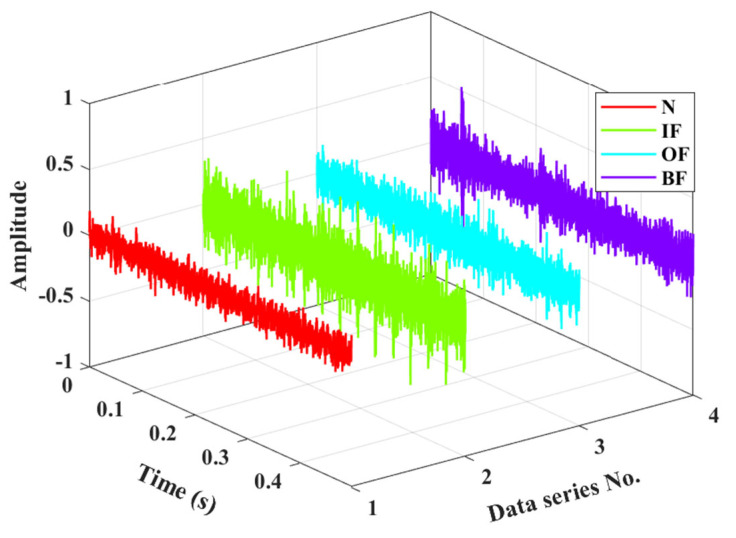
Vibration signals of bearings under different health conditions.

**Figure 4 entropy-27-00175-f004:**
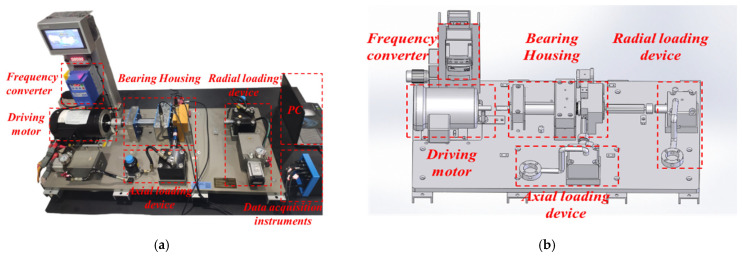
Rolling bearing fault diagnosis platform: (**a**) experimental testing system and (**b**) schematic diagram of the platform.

**Figure 5 entropy-27-00175-f005:**
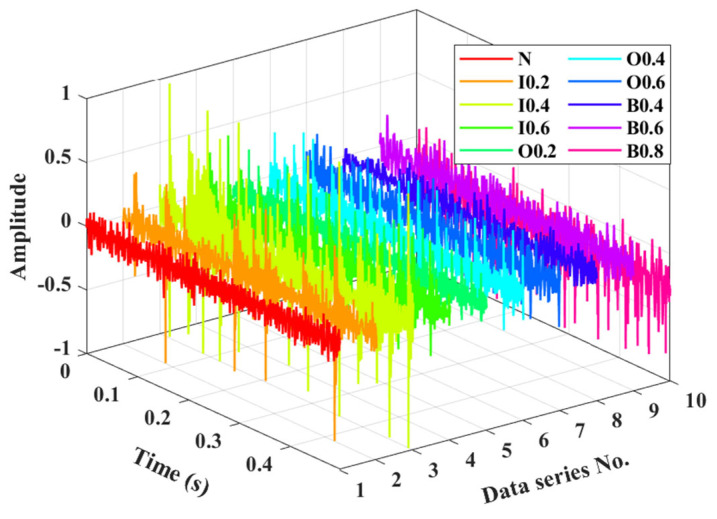
Vibration signals of bearings under different health conditions.

**Figure 6 entropy-27-00175-f006:**
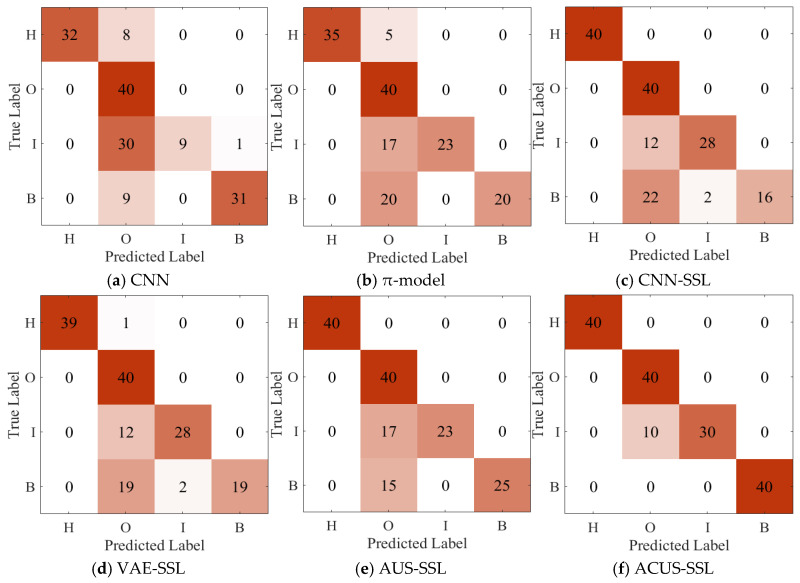
Confusion matrix for Experiment 1.

**Figure 7 entropy-27-00175-f007:**
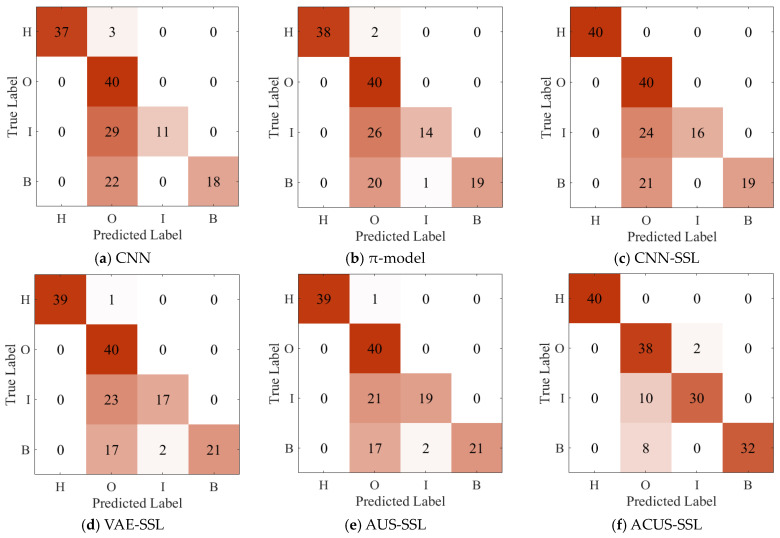
Confusion matrix for Experiment 4.

**Figure 8 entropy-27-00175-f008:**
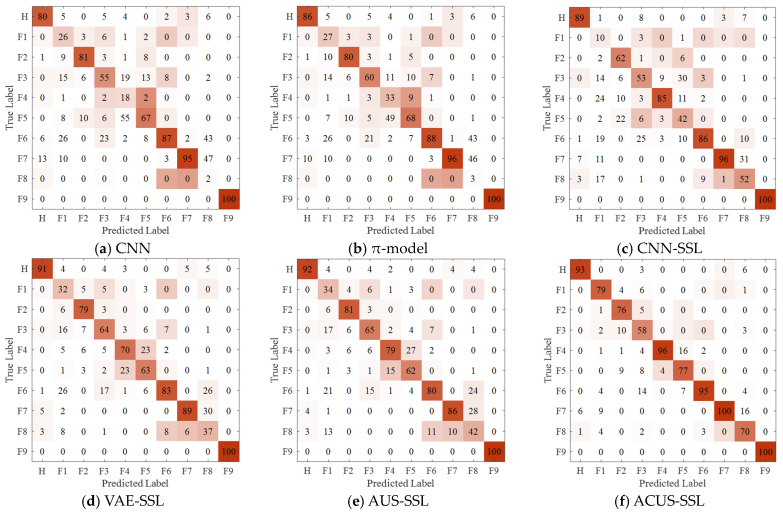
Confusion matrix for Experiment 7.

**Figure 9 entropy-27-00175-f009:**
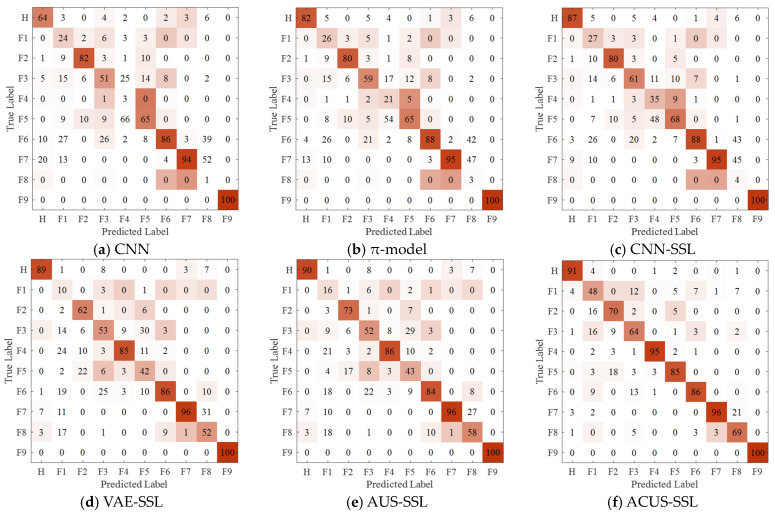
Confusion matrix for Experiment 10.

**Figure 10 entropy-27-00175-f010:**
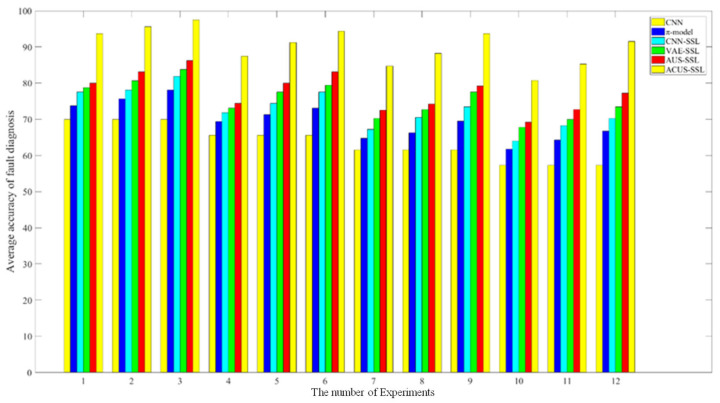
Comparison of the accuracy of all diagnostic models in 12 experimental scenarios.

**Table 1 entropy-27-00175-t001:** SKF6205 rolling bearing parameters.

Parameters	Pitch Diameter (mm)	Rolling Element Diameter (mm)	Number of Rolling Elements	Contact Angle (°)
Value	39.04	7.94	9	0

**Table 2 entropy-27-00175-t002:** Types of bearing faults in the CWRU datasets.

Fault Type	Fault Size (Inch)	Label
Normal	0	0
Inner fault	0.007	1
Outer fault	0.007	2
Ball fault	0.007	3

**Table 3 entropy-27-00175-t003:** ER-16K rolling bearing parameters.

Parameters	Inner Diameter (mm)	Outside Diameter (mm)	Number of Rolling Elements	Contact Angle ( °)
Value	25	52	9	0

**Table 4 entropy-27-00175-t004:** Types of bearing faults in the SMU datasets.

Fault Type	Fault Size (mm)	Label
Normal	0	H
Inner fault	0.2	F1
	0.4	F2
	0.6	F3
Outer fault	0.2	F4
	0.4	F5
	0.6	F6
Ball fault	0.4	F7
	0.6	F8
	0.8	F9

**Table 5 entropy-27-00175-t005:** Description of each model method.

Model	Method Description
CNN [2]	The CNN trained solely on labeled samples does not require unlabeled sample screening.
π-model [4]	The π-model leverages data augmentation techniques to perform consistency regularization loss learning on unlabeled data.
CNN-SSL [7]	The CNN-SSL achieves the labeling of unlabeled samples through joint fine-tuning of supervised and unsupervised models.
VAE-SSL [18]	The VAE-SSL utilizes VAE to learn the intrinsic features of the data, thereby improving the model’s ability to extract features from unlabeled samples.
AUS-SSL	The AUS-SSL employs active learning to screen high-quality unlabeled samples for use in supervised models.
ACUS-SSL	The ACUS-SSL designs a data imbalance-driven cost-sensitive function based on Model A to address the information imbalance issue encountered during the selection process of unlabeled samples.

**Table 6 entropy-27-00175-t006:** Experimental scenario settings and dataset construction.

Scenarios	Experiment	Number of Labeled Samples	Number of Unlabeled Samples	Number of Screened Unlabeled Samples
Scenario 1	Experiment 1	4 × 10	4000	50
	Experiment 2	4 × 10	6000	100
	Experiment 3	4 × 10	8000	200
Scenario 2	Experiment 4	4 × 5	4000	50
	Experiment 5	4 × 5	6000	100
	Experiment 6	4 × 5	8000	200

**Table 7 entropy-27-00175-t007:** Structure parameters of the proposed model.

Network Layer	Kernel Size	Stride	Number of Kernels/Neurons
Conv0	5 × 5	1	32
Pool0	2 × 2	2	--
Conv1	5 × 5	1	64
Pool1	2 × 2	2	--
Conv2	5 × 5	1	128
Pool2	2 × 2	2	--
FC1	--	--	256
FC2	--	--	128

**Table 8 entropy-27-00175-t008:** Diagnostic results of all methods on Experimental Scenario 1.

Experiment	Imbalance Ratio of Unlabeled Samples	CNN	π-Model	CNN-SSL	VAE-SSL	AUS-SSL	ACUS-SSL
Experiment 1	1:3:4:2	70%	73.75%	77.5%	78.75%	80.00%	**93.75%**
Experiment 2	1:13:4:2	70%	75.63%	78.13%	80.63%	83.13%	**95.63%**
Experiment 3	1:8:8:3	70%	78.13%	81.86%	83.75%	86.25%	**97.5%**

**Table 9 entropy-27-00175-t009:** Diagnostic results of all methods on Experimental Scenario 2.

Experiment	Imbalance Ratio of Unlabeled Samples	CNN	π-Model	CNN-SSL	VAE-SSL	AUS-SSL	ACUS-SSL
Experiment 4	1:3:4:2	65.63%	69.38%	71.88%	73.13%	74.38%	**87.50%**
Experiment 5	1:13:4:2	65.63%	71.25%	74.38%	77.50%	80.00%	**91.25%**
Experiment 6	1:8:8:3	65.63%	73.13%	77.50%	79.38%	83.13%	**94.38%**

**Table 10 entropy-27-00175-t010:** Experimental scenario settings and dataset construction.

Scenarios	Experiment	Number of Labeled Samples	Number of Unlabeled Samples	Number of Screened Unlabeled Samples
Scenario 3	Experiment 7	10 × 10	10,000	50
	Experiment 8	10 × 10	15,000	100
	Experiment 9	10 × 10	20,000	200
Scenario 4	Experiment 10	10 × 5	10,000	50
	Experiment 11	10 × 5	15,000	100
	Experiment 12	10 × 5	20,000	200

**Table 11 entropy-27-00175-t011:** Diagnostic results of all methods on Experimental Scenario 3.

Experiment	Imbalance Ratio of Unlabeled Samples	CNN	π-Model	CNN-SSL	VAE-SSL	AUS-SSL	ACUS-SSL
Experiment 7	1:3:7:3:4:9:10:5:1:7	61.1%	64.10%	67.5%	70.8%	72.1%	**84.40%**
Experiment 8	1:2:5:8:3:10:10:9:2	61.1%	66.25%	70.50%	72.75%	74.25%	**88.25%**
Experiment 9	1:7:13:8:3:5:2:3:4:4	61.1%	69.50%	73.50%	77.50%	79.25%	**93.75%**

**Table 12 entropy-27-00175-t012:** Diagnostic results of all methods on Experimental Scenario 4.

Experiment	Imbalance Ratio of Unlabeled Samples	CNN	π-Model	CNN-SSL	VAE-SSL	AUS-SSL	ACUS-SSL
Experiment 10	1:3:7:3:4:9:10:5:1:7	57%	61.9%	64.50%	67.5%	69.8%	**80.4%**
Experiment 11	1:2:5:8:3:10:10:9:2	57%	64.25%	68.25%	70.00%	72.75%	**85.25%**
Experiment 12	1:7:13:8:3:5:2:3:4:4	57%	66.75%	70.25%	73.50%	77.25%	**91.50%**

## Data Availability

The data involved in this article have been presented in the article.

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
