# Peer review of "Fault Diagnosis of Semi-Supervised Electromechanical Transmission Systems Under Imbalanced Unlabeled Sample Class Information Screening"

_entropy, 2025, doi:10.3390/e27020175_

Round 1

Reviewer 1 Report

Comments and Suggestions for Authors

This paper proposes a semi-supervised fault diagnosis method under the screening of imbalanced and unlabeled sample class information, and constructs multiple experimental scenarios on two bearing datasets to verify the effectiveness and superiority of the proposed method. Here are some suggestions for revisions to help improve the quality of this paper:

1. It is recommended that the authors correctly and consistently use the abbreviation throughout the manuscript after its first use.

2. The authors are encouraged to check the correctness of Eq.3. Additionally, it is recommended to verify the correctness of symbols in all formulas throughout this paper and ensure that each variable symbol has a corresponding explanation.

3. In Table 5, the literature citation for the VAE-SSL method is incorrect.

4. The authors are suggested to elaborate on the differences between the two bearing datasets.

5. Is the model method proposed in this paper applicable to fault diagnosis of electromechanical drive systems under variable speed conditions?

6. The authors are encouraged to explain what distinguishes the cost-sensitive function constructed in this paper from traditional methods.

7. It is suggested that the authors revise the citation format of the references according to the journal's requirements.

Reviewer 2 Report

Comments and Suggestions for Authors

This paper proposes a novel semi-supervised diagnostic method for accurate fault identification of rolling bearings in electromechanical drive systems under imbalanced unlabeled sample class information screening. The following are my specific comments:

1. In the flowchart of the proposed method shown in Figure 1, the authors mention loss_new, but it is not reflected in the corresponding theoretical section. It is recommended to check and make the necessary modifications.

2. Suggest the authors to increase the clarity of the images.

3. How was the unbalanced ratio of unlabeled samples obtained in Tables 8 and 9? Additionally, what is the purpose of using CNN for comparison in the comparative algorithms?

4. Please explain why the proposed method can achieve such high accuracy under noise interference.

5. In terms of conclusions, the author should provide a future prospect of such a network and possible improvements in the future.

6.The format of the references needs to be unified, some information is missing.

Reviewer 3 Report

Comments and Suggestions for Authors

The literature review is good. It covers major aspects of the existing state of the art in this research field. 

Figure 1 is excellent. It demonstrates the steps needed to optimize the CNN architecture before it can be used online for bearing fault diagnosis. 

The major question remains if this system is capable to adapt to different environmental and technological conditions. Bearings are different, as different are the machines where those bearings are used. How extensive (and expensive) is the offline training step depicted in Figure 1 for a new type of bearing (and a new type of machine)? Are there any possibilities to use transfer learning for carrying some parts of the optimized architecture from one bearing (machine) to another bearing (machine)?

Another question to the schematoc diagram in Figure 1. One of the blocks in the offline training part reads: "Comparison of expert experience knowledge and classification space prediction results". Well, the major driving force for intelligent fault prediction algorithms is usually based on the wish to eliminate the human operator induced errors and mistakes. Now, it appears that the CNN training is still dependent on the experience of human operators. The authors are requested to elaborate on those issues. 

Equation (6) is somewhat strange. Feature extraction from bearing vibration signal is the most imporatnt step in any fault diagnosis algorithm. Dropping a single equation without further explanations somehow lessens the interest to this manuscript. A typical example explaining one particular algorithm for the feature extraction is presented in [A]. The authors are at least requested to comment on [A]. 

The authors are using log_2 for the nomalization of diff (Equation (8)). Why log_2, but not log_10 or ln? What are the requirements for the constant C?

The authors do use the gradient of loss in Equation (16). Why not conjugate gradients? It is well known that conjugate gradients may outperform the gradient descent in many practical applications. 

[A] M.Landauskas, M.Cao, M.Ragulskis. Permutation entropy based 2D feature extraction for bearing fault diagnosis. Nonlinear Dynamics (2020) vol.102, 1717-1731.
